The role of temperature on the development of circadian rhythms in honey bee workers

Giannoni-Guzmán Manuel A. 1 manuel.giannoni10@gmail.com
Perez Claudio Eddie 2
Aleman-Rios Janpierre 3
Diaz Hernandez Gabriel 3
Perez Torres Melina 4
Melendez Moreno Alexander 3
Loubriel Darimar 3
Moore Darrell 5
Giray Tugrul 3
Agosto-Rivera Jose L. 3
1 Department of Biological Sciences, Vanderbilt University , Nashville, Tennessee , United States
2 Department of Biomedical Informatics, University of Pittsburgh , Pittsburgh, Pennsylvania , United States
3 Department of Biology, University of Puerto Rico Rio Piedras , San Juan, Puerto Rico , United States
4 Department of Biology, Brandeis University , Waltham, Massachusetts , United States
5 Department of Biological Sciences, East Tennessee State University , Johnson City, Tennessee , United States
Reser David
Electronic publication date: 2024 Mar 15
Publication date: 2024
Volume: 12
Electronic Location ID: e17086
Received 2023 Sep 19; Accepted 2024 Feb 20
Copyright: © 2024 Giannoni-Guzmán et al.
Copyright year: 2024
Copyright holder: Giannoni Guzmán et al.
License: This is an open access article distributed under the terms of the Creative Commons Attribution License, which permits unrestricted use, distribution, reproduction and adaptation in any medium and for any purpose provided that it is properly attributed. For attribution, the original author(s), title, publication source (PeerJ) and either DOI or URL of the article must be cited.
License URL: https://creativecommons.org/licenses/by/4.0/

Keywords: Circadian rhythms, Honey bees, Workers, Temperature, Development

Funding: National Science Foundation (NSF) awards 1026560, 1633184, 1707355, 2231637 National Institute of Health (NIH) 2R25GM061151-13, P20GM103475 This work was sponsored by the National Science Foundation (NSF) awards 1026560, 1633184, 1707355, 2231637 and the National Institute of Health (NIH) 2R25GM061151-13, P20GM103475. The funders had no role in study design, data collection and analysis, decision to publish, or preparation of the manuscript.

==============================
Circadian rhythms in honey bees are involved in various processes that impact colony survival. For example, young nurses take care of the brood constantly throughout the day and lack circadian rhythms. At the same time, foragers use the circadian clock to remember and predict food availability in subsequent days. Previous studies exploring the ontogeny of circadian rhythms of workers showed that the onset of rhythms is faster in the colony environment (~2 days) than if workers were immediately isolated after eclosion (7–9 days). However, which specific environmental factors influenced the early development of worker circadian rhythms remained unknown. We hypothesized that brood nest temperature plays a key role in the development of circadian rhythmicity in young workers. Our results show that young workers kept at brood nest-like temperatures (33–35 °C) in the laboratory develop circadian rhythms faster and in greater proportion than bees kept at lower temperatures (24–26 °C). In addition, we examined if the effect of colony temperature during the first 48 h after emergence is sufficient to increase the rate and proportion of development of circadian rhythmicity. We observed that twice as many individuals exposed to 35 °C during the first 48 h developed circadian rhythms compared to individuals kept at 25 °C, suggesting a critical developmental period where brood nest temperatures are important for the development of the circadian system. Together, our findings show that temperature, which is socially regulated inside the hive, is a key factor that influences the ontogeny of circadian rhythmicity of workers.

Introduction

Honey bee colonies can efficiently regulate the colony microenvironment. In the brood-rearing seasons (spring and summer), honey bee colonies socially regulate colony temperature, keeping the brood nest optimally at 35 °C (Simpson, 1961; Kronenberg & Heller, 1982; Jones et al., 2004; Jones, Nanork & Oldroyd, 2007; Heinrich, 1980). Thermoregulation of honey bee colonies is essential for colony performance and survival (Heinrich, 1981, 1993). Experiments examining the effects of low temperatures (28–30 °C) on brood development show that these temperatures can cause malformations in the brood, while brood kept at high temperatures (38–40 °C) exhibit high mortality rates (Himmer, 1927; Heinrich, 1993). Subsequent studies showed that pupal development at 32 °C, only 3 °C lower than optimal core temperature, results in significant deficits in waggle dance behavior and learning and memory assays, compared to bees raised at 36 °C (Tautz et al., 2003). However, little is known about the effects of colony temperature on the development of circadian locomotor rhythms in honey bees.

The circadian clock of honey bees is essential in complex physiological processes, such as spatiotemporal learning, time perception, and sun-compass navigation (von Frisch, 1967; Goodwin & Lewis, 1987; Moore et al., 1998; Van Nest & Moore, 2012; Wagner et al., 2013). However, when it comes to the development of circadian rhythms in honey bee workers, scientists are just beginning to scratch the surface of what is thought to be a highly complex mechanism of regulation, with components at the environmental, social, hormonal, and genetic levels (Moore et al., 1998; Moore, 2001; Shemesh, Cohen & Bloch, 2007; Eban-Rothschild & Bloch, 2012).

The development of honey bee circadian rhythms is of particular interest because, similar to human infants, young honey bees present postembryonic development of circadian rhythms at the behavioral, molecular, and neuroanatomical levels (Moore et al., 1998; Bloch, Toma & Robinson, 2001; Eban-Rothschild, Shemesh & Bloch, 2012; Fuchikawa et al., 2017; Beer & Helfrich-Förster, 2020). Studies examining the timing of in-hive tasks such as brood care found that individual ‘nurses’ performed this task around the clock, which is thought to benefit the developing brood (Moore et al., 1998). Furthermore, in the colony, workers will remain arrhythmic, performing in-hive tasks and developing circadian rhythmicity just before the onset of foraging behavior, indicating that the ontogeny of circadian rhythms is intertwined with the age-related division of labor in the colony.

In isolation, during the first days of their adult life, young bees lack behavioral, metabolic, or daily oscillations in circadian gene expression in the brain associated with circadian rhythmicity. Under these constant conditions (DD, ~60%RH, 26–30 °C), researchers have reported that onset of circadian rhythmicity occurs at around 7–10 days of age in 50% of the sampled subjects (Toma et al., 2000; Moore, 2001). Furthermore, under these experimental conditions, by 16 days of age, around 25% of the bees were still arrhythmic (Toma et al., 2000). Since the ontogeny of circadian rhythms is thought to be regulated by age-related division of labor, researchers have manipulated neuroendocrine signals known to accelerate the onset of foraging (such as juvenile hormone, octopamine, and cGMP-dependent protein kinase), hypothesizing a similar effect on circadian rhythms but finding no effect in individually isolated bees (Bloch, Sullivan & Robinson, 2002; Ben-Shahar, 2003; Meshi & Bloch, 2007). Examining the activity patterns inside the colony in the presence or absence of brood revealed that brood inhibits the circadian rhythms of young workers in the colony (Shemesh et al., 2010). Another study by the same group tested whether the colony environment or other social cues may elicit strong circadian rhythms in young workers (Eban-Rothschild, Shemesh & Bloch, 2012). Their findings reveal that experiencing the colony environment in a mesh cage for 48 h after adult emergence resulted in a significantly greater number of bees presenting circadian rhythms in the laboratory, compared to isolated and group caged bees. The authors of this work postulate that social cues, the colony microenvironment, or a combination of both play a role in the ontogeny of circadian rhythms of young workers. Taken together, these studies suggest the existence of a cue, or combination of signals, that elicits the development of circadian rhythmicity.

Environmental temperature is also important for locomotor activity rhythms. Studies examining the endogenous rhythms of the Japanese honey bee Apis cerana show that environmental temperature directly affects the endogenous circadian period length of foragers (Fuchikawa & Shimizu, 2007). Consistent with the previous study, work in our laboratory using the gentle Africanized honey bee (gAHB) also shows that increasing environmental temperature lengthens the endogenous period length in honey bee foragers (Giannoni-Guzmán et al., 2014). Furthermore, oscillations observed inside bee colonies are sufficient to entrain the circadian clock of foragers in the absence of other time givers (Giannoni-Guzmán et al., 2021). However, the effect of temperature on the development of circadian rhythms in honey bee workers has yet to be explored.

In the current study, we examined the effects of environmental temperature on the development of circadian rhythms in young workers. We hypothesized that temperature at the brood-nest plays an important role in the development of circadian rhythms in young honey bee workers. To test this hypothesis, we isolated 1-day-old workers in locomotor activity monitors at low (24–26 °C) or brood-nest-like (33–35 °C) temperatures. We quantified the proportion of rhythmic individuals in each group and compared the endogenous period length of rhythmic individuals and the mortality within and between groups. Furthermore, to determine if brood-nest-like temperature’s effects have a developmental time window (critical period), we examined the impact of brood-nest-like temperature for the first 48 h of our paradigm. Our results highlight the importance of the socially regulated temperature of the hive in the ontogeny of circadian rhythms in honey bee workers.

Materials and Methods

Honey bee colonies and worker collections

Portions of this article were previously published as part of a preprint (https://doi.org/10.1101/2020.08.17.254557). Colonies used in our experiments had mated queens laying eggs of gentle Africanized honey bees (Galindo-Cardona et al., 2013). These colonies were located at the University of Puerto Rico (UPR) Gurabo Experimental Station in Gurabo, Puerto Rico. For all experiments, brood frames were collected, workers were removed, and then the frame was stored in an incubator overnight (~35 °C). The following morning, bees that emerged from the frames were collected and placed inside individual tubes for locomotor activity monitoring. This process briefly exposed the bees to light while transferring from the colony frame to the individual tubes. The first colony of experiment 1 was examined on November 29, 2012 (colony 1), while the second colony was assayed beginning January 12, 2013 (colony 2). A total of 384 bees were used in this experiment, 128 for colony 1 and 256 for colony 2. Experiment 2 examined the effect of temperature during the first 48 h after eclosion on the development of circadian rhythms beginning on February 26, 2016.

Experiment 1: development of circadian rhythms at low (24–26 °C) vs high (33–35 °C) temperatures

Locomotor activity measurements were carried out using two environmental chambers (Percival, I-30BLL) set up under constant darkness and relative humidity of 60 ± 5%, either at low temperature (24–26 °C) or brood nest-like temperatures (33–35 °C). Locomotor activity was recorded using monitors and software from Trikinetics (Waltham, MA, USA) as previously described (Giannoni-Guzmán et al., 2014). Briefly, 1-day-old workers were collected from the brood frame and placed inside individual tubes within the activity monitoring system. Food in the form of honey candy (mixed sugar and honey) and water were provided “ad-libitum” and changed as needed under dim red light. Circadian rhythmicity was determined using 4 consecutive days of data (days 6–10), using autocorrelation analysis for 1-min bins (Levine et al., 2002). All bees were approximately the same age for periods where rhythmicity was analyzed.

Experiment 2: development of circadian rhythms after 48 h at 35 °C

As in experiment 1, we measured locomotor activity using two environmental chambers. In one of these, the temperature during the first 48 h was set at 35 °C and afterward lowered to 25 °C for the remainder of the experiment. The other incubator was kept at 25 °C throughout the experiment. Food and water were provided ad libitum and changed as needed under dim red light.

Data analysis

We used the freely available software Circadian Dynamics (https://github.com/edpclau/circadian-dynamics) to study rhythmic patterns in the locomotor activity of each individual. Circadian Dynamics applies classic rhythm analysis methods in a constrained sliding window. This approach allows users to assess how rhythmic dynamics change over time for each individual. For our analysis, we set the sliding window to span 4 days with a step size of 1 day. For example, if the first window starts on Sunday, the analysis will cover all the data produced on Sunday, Monday, Tuesday, and Wednesday. The second window will cover all the data produced on Monday, Tuesday, Wednesday, and Thursday.

Circadian rhythmicity was determined through statistical analysis and confirmed by visual inspection of activity plots. Visual inspection of the data for those windows and of the whole actogram was utilized to confirm the statistical results. The endogenous period length and rhythmicity for each window was determined using a combination of autocorrelation (Levine et al., 2002) and the Lomb-Scargle periodogram (Ruf, 1999). To examine statistical differences in average period length across treatment groups, we performed unpaired t-tests for each colony. The age of onset of circadian rhythms was determined by the first day of a three-window sequence, where the individual was consistently evaluated to be rhythmic. The actograms of each individual were plotted using freely available MATLAB® toolboxes developed in Jeffrey Hall’s laboratory (Levine et al., 2002).

To determine differences in the proportion of rhythmic bees in our experiments we used generalized estimating equations, which account for the longitudinal nature of our data set (Halekoh, Højsgaard & Yan, 2006). We performed survival analysis via the Gehan-Breslow-Wilcoxon test to determine if environmental temperature influences survival in our experiments. A two-way ANOVA was performed for each experiment to examine differences in survival based on the development of rhythms within and between cohorts. All statistical analyses were performed using the R-Studio and R. Graphs and figures were created in MATLAB (MathWorks, Inc., Natick, MA, USA) and GraphPad Prism 9.00 (GraphPad Software, La Jolla, CA, USA). The code and raw data to replicate our analyses are available at 10.5281/zenodo.10519457.

Results

Young workers kept at higher temperatures (33–35 °C) developed circadian rhythms as early as 2 days of age compared to young workers kept at lower temperatures (24–26 °C), which began developing rhythms between 4–5 days of age (Fig. 1). In addition, 60–80% of workers developed circadian rhythms at brood nest-like temperatures. At lower temperatures, less than 40% of the bees developed rhythmicity (generalized estimating equations, Colony 1: time p < 0.001, temperature p < 0.001, interaction p < 0.001; Colony 2: time p < 0.001, temperature p < 0.001, interaction p < 0.001) (Fig. 1). This result is in line with our hypothesis that brood nest temperature influences the ontogeny of circadian rhythms.

Figure 1 The rate and proportion of young workers developing circadian rhythms are greater at high (33–35 °C) than at lower (24–26 °C) temperatures.

Cumulative distributions of rhythmic young workers at low and high temperatures in constant darkness for two colonies: (A) Colony 1, (B) Colony 2. Dotted line across each plot represents 0.5 proportion of rhythmic bees. At high temperatures (33–35 °C), the rate of development and the proportion of bees developing strong circadian rhythms were significantly higher than at lower temperatures, as shown by generalized estimating equations.

Previous work on different species of honey bees has shown that environmental temperature affects the endogenous period length of forager rhythms (Fuchikawa & Shimizu, 2007; Giannoni-Guzmán et al., 2014). We hypothesized that rhythmic young workers would present endogenous rhythms closer to 24 h when assayed at temperatures closer to those of the brood nest than those at lower temperatures. Consistent with previous work (Spangler, 1972; Moore & Rankin, 1985; Toma et al., 2000; Giannoni-Guzmán et al., 2014), our results in two different colonies (Fig. 2) show that, on days 6–10 of the experiment, bees kept at lower temperatures exhibited significantly shorter mean endogenous period lengths (23.10 ± 0.29 h SEM) than bees in brood nest like temperatures (24.5 ± 0.13 h SEM) (unpaired t-test Colony 1: t(40) = 3.835, p < 0.001; unpaired t-test Colony 2: t(81) = 4.964, p < 0.001).

Figure 2 Locomotor activity patterns of young honey bee workers under high or low-temperature constant darkness.

Double-plotted actograms of representative workers at (A) low and (B) high temperatures in constant darkness. Autocorrelation plots were used to determine the rhythmicity of locomotor activity and calculate the endogenous period length (p), rhythm index (RI), and rhythm strength (RS) from days 1–5 and 6–10 for each individual. (C and D) The mean free-running period for rhythmic individuals at high temperatures (24.5 ± 0.13 h SEM) was closer to 24 h and significantly different from that measured at low temperatures for both colonies (23.10 ± 0.29 h SEM) (unpaired t-test Colony 1: t(40) = 3.835, p = 0.0004***) (unpaired t-test Colony 2: t(81) = 4.964, p < 0.0001****).

The low and high-temperature cohorts also differed markedly with respect to mortality levels. By day 10, only ~30% of bees from the low-temperature groups survived. In comparison, the mortality of the higher temperature groups was significantly lower, with more than 65% of the bees still alive at the end of each experiment (Gehan-Breslow-Wilcoxon test, p < 0.001). Furthermore, by separating each cohort by individuals who did or did not develop circadian rhythms, we observed a relationship between rhythmicity and mortality (Figs. 3C and 3D). Two-way ANOVA revealed significant effects for environmental temperature, rhythmicity status, and their interaction (Colony 1: rhythmicity p < 0.001, temperature p < 0.001, interaction p < 0.03; Colony 2: rhythmicity p < 0.001, temperature p < 0.001, interaction p = 0.99). Post hoc analysis using Šídák’s multiple comparisons tests was performed to compare differences between and within groups. Significant differences in the survival of arrhythmic and rhythmic bees at low temperatures and between arrhythmic individuals in low and high temperatures were observed for both colonies. Comparing the mean survival of rhythmic individuals in low and high temperatures resulted in significance only for colony 2. Similarly, our post hoc analysis found significant differences within the high-temperature group for Colony 2 but not Colony 1. These could result from colony-to-colony differences and a difference in the number of individuals assayed. Overall, these differences in mortality suggest that the development of circadian rhythms is correlated with the survival of young bees in both low temperatures and those similar to the brood nest.

Figure 3 Survival of isolated young workers decreases at lower temperatures and in arrhythmic individuals.

Survival plot of 1-day-old honey bee cohorts of (A) Colony 1 and (B) Colony 2 at low (solid line) and high (intermittent line) temperature conditions. Statistical comparisons of the cohorts revealed that the survival of individuals was higher in the high-temperature cohort (Gehan-Breslow-Wilcoxon, p < 0.0001****). Panels (C and D) show bar graphs of mean survival and standard error for arrhythmic and rhythmic individuals separated by experimental cohort (low or high). Asterisks indicate significant differences in Šídák’s multiple comparisons tests (Adjusted p < 0.05*).

Our findings strongly suggest that temperature positively influences the rate and proportion of young worker bees developing circadian rhythms (Fig. 1). Intriguingly, Eban-Rothschild, Shemesh & Bloch (2012) showed that the first 48 h of adult life in the colony influences the development of strong circadian rhythms. Combining these two discoveries, we hypothesize a critical period (the first 48 h after adult emergence) for the influence of temperature on the development of circadian rhythmicity in worker honey bees. If colony temperature is, indeed, a crucial factor in the development of circadian rhythmicity in young workers, then placing 1-day-old workers at 35 °C during the first 48 h after emergence but afterward keeping the environmental temperature at 25 °C will result in a greater proportion of workers developing circadian rhythms than 1-day old bees continuously subjected to 25 °C conditions. Consistent with this hypothesis, we found that exposure to 35 °C during the first 48 h after emergence plays a significant role in the development of circadian rhythms in young workers (Fig. 4A). The cumulative proportion of rhythmic individuals for the 35–25 °C group was significantly greater than that in the 25 °C group (generalized estimating equations, time p < 0.001, temperature p < 0.001, interaction p < 0.001). In addition to the effects of temperature on the development of circadian rhythms, we also observed significant differences in the survival of individuals exposed to 35 °C for the first 48 h and those kept at 25 °C. By day seven, less than 10% individuals had died in the 35–25 °C group, while more than 40% had died in the 25 °C (Gehan-Breslow-Wilcoxon, n = 256, p < 0.001). Taken together, our results indicate that the development of circadian rhythms in workers exhibits a temperature-sensitive critical period.

Figure 4 Temperature (35 °C) during the first 48 h after emergence increases the rate and number of rhythmic workers.

(A) Cumulative distribution of rhythmic young workers exposed to 35 °C during the first 48 h after emergence and afterward placed at 25 °C for the remainder of the experiment (dashed line) compared to that of bees placed at 25 °C (solid line) after emergence. Generalized Estimating Equations revealed significant differences between the rate and proportion of individuals developing rhythmic behavior under these conditions (Time p < 0.001***, Temperature p < 0.001***, Interaction p < 0.001***). (B) Survival plot of 1-day-old honey bee cohorts at 25 °C (solid line) and bees exposed to 35 °C for the first 48 h after emergence (intermittent line). Individuals in the 35–25 °C cohort presented significantly better survival rates than bees placed at 25 °C since the beginning of the experiment (Gehan-Breslow-Wilcoxon, n = 256, p < 0.001***).

Discussion

Previous studies exploring the ontogeny of circadian rhythms of workers showed that the onset of rhythms is faster in the colony environment (~2 days) than if workers were immediately isolated after eclosion (7–9 days) (Toma et al., 2000; Eban-Rothschild, Shemesh & Bloch, 2012; Giannoni-Guzmán et al., 2014). Specifically, previous work has shown that 60–80% of bees taken from the hive 48 h after eclosion presented strong locomotor rhythms as soon as they were in isolation (Eban-Rothschild, Shemesh & Bloch, 2012). Additionally, a set of experiments by Shemesh et al. (2010) showed that direct interactions with other colony members or the brood could not account for this difference. However, which specific environmental factors influenced the early development of worker circadian rhythms remained unknown. Here, we add to this body of work by showing that brood nest temperature in the absence of all other colony cues is a key factor determining the development of circadian rhythms (Fig. 1). Our work is consistent with previous studies that examined the developmental rate of rhythms at 26 °C. In these studies up to 50% of workers developed rhythms by day 12 (Toma et al., 2000; Moore, 2001).

Furthermore, we show that keeping young workers at 35 °C for the first 48 h after eclosion is sufficient to elicit early rhythm development. This suggests a critical window in which brood nest temperature influences the development of circadian rhythms in honey bees (Fig. 4). In addition, we find that survivorship between and within experimental groups is positively associated with the presence of locomotor rhythms (Fig. 3). In sum, our results strongly indicate that brood nest temperature plays a crucial role in the development of circadian rhythms in honey bee workers.

In the present study, exposure to brood nest-like temperatures during the first 48 h after emergence resulted in twice the proportion (40%) of rhythmic bees than bees at 25 °C (20%). Furthermore, our results in the 25 °C groups are similar to those of bees that only spent the first 24 h after eclosion inside the colony and were then placed in locomotor activity monitors at 29 °C in a previous study (Eban-Rothschild, Shemesh & Bloch, 2012). Based on these results, we hypothesize that 24 h of exposure to the colony environment is not sufficient to elicit the development of the circadian clock. Thus, our results and those from previous work indicate the presence of a critical window, as defined by Sengpiel (2007), where brood nest temperature impacts the development of circadian rhythms in honey bee workers. To better define the extent of this critical period in the future we can vary the timing and number of continuous hours of 35 °C.

Concerning the survival rates observed in bees at brood nest-like and low temperatures (Fig. 3), our analyses revealed that environmental temperature and lack of rhythmicity act independently to reduce survival rates. The effect of temperature on survival is consistent with previous work showing that changes in temperature during development can have long-lasting effects on the rate of behavioral development and dance performance (Tautz et al., 2003; Becher, Scharpenberg & Moritz, 2009). It is possible that in addition to the development of circadian rhythms, other systems are still under development and do not develop properly at low temperatures, causing the observed mortality.

The connectivity among various systems is necessary to elicit circadian rhythms of locomotion. At the brain level, in the fruit fly, it is known that multiple oscillators that control the timing of locomotor activity at different times of the day (e.g., morning and evening cells) not only need to communicate but they need to synchronize in a specific manner (Stoleru et al., 2004, 2005). Recent work in the honey bee tracking the number of Pigment Dispersing Factor (PDF) positive neurons across different ages in the worker has shown that the number of PDF+ neurons and their arborizations to different parts of the bee brain continues to change after emergence (Beer & Helfrich-Förster, 2020; Beer, Härtel & Helfrich-Förster, 2022). It is thus possible that the establishment of connections among the multiple components of the circadian system in the honey bee brain occurs in the first 48 h after emergence. Alternatively, perhaps the development of functional connections between clock neurons in the brain and central pattern generators (CPGs) responsible for driving locomotor activity located elsewhere in the central nervous system, are regulated by temperature soon after emergence. Without synchronizing signals from the brain, initiating and regulating locomotor rhythms may be impossible: arrhythmicity might be explained by the failure to establish these connections.

Our study has its limits. One of the limitations is that bees were only fed honey candy during our experiments. Pollen which is essential for honey bee health and survival was not provided (Brodschneider & Crailsheim, 2010; Di Pasquale et al., 2016). Previous work like the Toma et al. (2000) fed the bees sugar syrup and honey and their results at 26 °C are similar to the low temperature results (Fig. 1) shown here. While our design did not include a group of bees inside the colony, previous studies have shown how the colony environment can influence the development of circadian rhythms of workers showing that 2 days inside of the colony is sufficient to elicit greater development of rhythms when compared to bees isolated after eclosion (Eban-Rothschild, Shemesh & Bloch, 2012; Beer & Helfrich-Förster, 2020).

While the data shown here provides strong evidence for a critical window where temperature influences rhythm development in young workers, the exact length of this window remains to be elucidated. Taking into consideration our work and work from other laboratories, we can hypothesize that the window begins within the first 48 h after emergence (Eban-Rothschild, Shemesh & Bloch, 2012; Beer & Helfrich-Förster, 2020). Future experiments combining locomotor and neuroanatomical measurements could establish how many days of 35 °C temperature are needed to fully mature the circadian system in worker bees. Additionally, further work is still needed to examine the importance of temperature on the development of circadian rhythms relative to other factors, such as genetic background and social cues inside the colony. Carefully examining the changes at the neural and gene expression levels occurring during the first 48 h of adult life may provide further insights into the mechanisms driving the ontogeny of circadian rhythms in honey bee workers.

We want to thank Dr. Arian Avalos and Dr. Emmanuel Rivera for help with the experiments, Dr. Arian Avalos and Dr. Luis de Jesus for their comments and suggestions. We would also like to recognize the director, Manuel Diaz, and the personnel of the Gurabo Experimental Agriculture Station of the University of Puerto Rico at Mayagüez for the use of facilities at “Casa Amarilla”.

Additional Information and Declarations

Competing Interests

Author Contributions

Data Availability

Tugrul Giray is an Academic Editor for PeerJ. The authors declare no other competing interests.

Manuel A. Giannoni-Guzmán conceived and designed the experiments, performed the experiments, analyzed the data, prepared figures and/or tables, authored or reviewed drafts of the article, and approved the final draft.

Eddie Perez Claudio analyzed the data, prepared figures and/or tables, authored or reviewed drafts of the article, and approved the final draft.

Janpierre Aleman-Rios performed the experiments, prepared figures and/or tables, and approved the final draft.

Gabriel Diaz Hernandez performed the experiments, prepared figures and/or tables, and approved the final draft.

Melina Perez Torres performed the experiments, prepared figures and/or tables, and approved the final draft.

Alexander Melendez Moreno performed the experiments, prepared figures and/or tables, and approved the final draft.

Darimar Loubriel performed the experiments, prepared figures and/or tables, and approved the final draft.

Darrell Moore conceived and designed the experiments, prepared figures and/or tables, authored or reviewed drafts of the article, and approved the final draft.

Tugrul Giray conceived and designed the experiments, authored or reviewed drafts of the article, and approved the final draft.

Jose L. Agosto-Rivera conceived and designed the experiments, authored or reviewed drafts of the article, and approved the final draft.

The following information was supplied regarding data availability:

The data and scripts are available at GitHub and Zenodo:

- https://github.com/giannoma/Ontogeny-Bee/tree/DOI.

- Manuel A. Giannoni-Guzmán. (2024). giannoma/Ontogeny-Bee: Zenodo Archive (DOI). Zenodo. https://doi.org/10.5281/zenodo.10519457.

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
