# Peer review of "The role of temperature on the development of circadian rhythms in honey bee workers"

_PeerJ, doi:10.7717/peerj.17086_

## Round 0.1 · original submission · Major Revisions

Both reviewers were largely positive in their comments, and I believe the suggestions they have put forward will improve the final manuscript. Please address each of their comments in your revised version, and I look forward to reading the next submission. Thank you for contributing your work to PeerJ.

Reviewer 1 ·

Basic reporting

The authors Giannoni-Guzmán et al. present a study on rhythmic behavior of young honey bees, in which they investigate rhythm development after emergence of the bees dependent on the experimental temperature. They find that brood nest temperatures improve speed and rate of circadian rhythm development compared to experiments in lower temperatures. Survival rate is higher in experiments conducted at brood nest temperatures. Furthermore, exposure to brood nest temperature for 48 hours after emergence seems to be sufficient to trigger these effects.
The study is well conducted and explained and discussed in detail. Statistics are sound and references are fitting. However, I have a few remarks that may help to improve the manuscript.

Experimental design

Major:
Experiment 2: To speak of a real rescue of behavior the control experiment of 35°C-35°C is missing. It may be that you find some minor effect compared to bees kept at 25°C after 48 hours at 35°C. And the temperature effect may still be there in older bees, but gradually fading.
In this context of gradual effects: You should compare with studies, in which circadian rhythm development has been observed under constant intermediate temperatures between the 25°C and 35°C temperatures, you have chosen. Do percentages of rhythmic bees and speed of circadian rhythm development in these studies also lie between the results of your two temperature treatments?

Validity of the findings

no comment

Additional comments

Minor comments:
Figure 3: Sample sizes in the graph would help
L85: A citation is missing
L92: “young workers”
L95: “interacting with other bees for 48 hours after adult emergence resulted in strong circadian rhythms” Was that the case for all types of interactions?
L146: Young bees need pollen to develop properly. Did you provide them with pollen?
L180: “environmental temperature”
L181: Did you mean “survival”?

Reviewer 2 ·

Basic reporting

no comment

Experimental design

no comment

Validity of the findings

no comment

Additional comments

In this manuscript, Guzman et al. characterized the effects of temperature on the development of circadian rhythms in honeybees. The authors demonstrate that an ambient temperature range of 33-35°C is conducive to the proper development of these biological rhythms. In contrast, lower temperatures were found to negatively impact not only the development of circadian rhythms but also the overall survival rates of the bees. The findings are compelling, and the experimental methodology appears to be sound. I do, however, have a few minor suggestions for improvement.

Abstract:
Line 6: ‘If’ should be ‘if’.


Results and Figures:
1. The authors begin the results section with a summary statement: "Our results are consistent with the hypothesis that brood nest temperature influences the ontogeny of circadian rhythms." This approach is not ideal. It would be more appropriate to first present the data and findings, and then discuss their implications in relation to the hypothesis.
2. It would have been nice to have a control group of bees inside the hive (positive control). I realize that these experiments were conducted about a decade ago, so it is impossible to add this now. It would be good to mention the lack of this control and the expected rate of development in hive bees.
3. Proportion data is not normally distributed and thus cannot be analyzed using parametric tests (such as ANOVA). The authors should use either Chi-Square Tests (but these do not account for the longitudinal nature of the data) or Generalized Estimating Equations.
4. General comment, please clearly indicate what is shown in each Panel of the figures (A, B etc).
5. Figure 1 – what does the dashed line represent?

---

## Round 0.2 · accepted · Accept

Thank you for attending carefully to the comments provided by the reviewers. I believe that their feedback and your amendments have yielded significant improvements in the manuscript, and I look forward to seeing the final version in print. Please consider Reviewer 1's comment regarding the 'rescue' terminology in Figure 4.

Reviewer 1 ·

Basic reporting

I consider the changes to the manuscript satisfactory and recommend the manuscript for publication. However, I still strongly advice the authors to not speak of a rescue in behavior (figure 4). It is technically wrong to speak of a “rescue”, when the control experiment is missing.

Experimental design

no comment

Validity of the findings

no comment

Reviewer 2 ·

Basic reporting

NA

Experimental design

NA

Validity of the findings

NA

Additional comments

The authors have addressed all my concerns.